# Anytime Depth Estimation with Limited Sensing and Computation Capabilities on Mobile Devices

**Yuedong Yang**      **Zihui Xue**      **Radu Marculescu**
The University of Texas at Austin
{albertyoung, sherryxue, radum}@utexas.edu

**Abstract:** Depth estimation is a safety critical and energy sensitive method for environment sensing. However, in real applications, the depth estimation may be halted at any time, due to the random interruptions or low energy capacity of battery when using powerful sensors like 3D LiDAR. To address this problem, we propose a depth estimation method that is robust to random halts and relies on energy-saving 2D LiDAR and a monocular camera. To this end, we formulate the depth estimation as an anytime problem and propose a new metric to evaluate its robustness under random interruptions. Our final model has only 2M parameters with a marginal accuracy loss compared to state-of-the-art baselines. Indeed, our experiments on NYU Depth v2 dataset show that our model is capable of processing 224×224 resolution images and 2D point clouds with any computation budget larger than 6.37ms (157 FPS) and 0.2J on an NVIDIA Jetson TX2 system. Evaluations on KITTI dataset under supervised and self-supervised training show similar results.

**Keywords:** Depth Estimation, Anytime Algorithm, Energy-aware Optimization, Mobile Devices

## 1   Introduction

Depth estimation plays a critical role in various applications, such as autonomous driving, augmented reality and virtual reality. While attempting to measure the distance from camera to obstacles, existing approaches resort to depth sensors like LiDAR or structured-light sensors. The goal is to reconstruct the depth in full resolution utilizing images, along with sparse depth measurements.

A variety of model architectures [1–12] have been proposed to effectively fuse the RGB images and LiDAR point clouds (PC). However, most approaches *cannot* be directly deployed on small robot platforms or mobile devices, say a mobile phone or AR glasses. This is due to multiple factors: First, these methods are passive to the changes in the environment. In contrast, a dynamic behavior is a must for a real robotic system. For example, when an obstacle appears suddenly, it is preferably to sacrifice accuracy for a fast response to avoid crashing. Second, many methods rely on bulky, heavy and high-end LiDAR systems that consume much power to generate dense and accurate PC [13–15]. On the contrary, depth sensors installed on edge devices may produce extremely sparse and non-uniformly distributed depth patterns, *e.g.*, a depth line produced by a cheap 2D LiDAR as shown in Figure 1. The sparsity pattern generated by a 2D LiDAR is very regular such that the PC can be represented with an 1D vector, while other types of PC in Figure 1(b) can only be represented with high-dimensional arrays. Moreover, to boost performance, deeper and more complicated models are adopted together with computation-intensive algorithms [5, 16]. This usually requires powerful GPU clusters and those are not available in robotic systems based on edge devices.

In this work, we argue for a more realistic problem setting for edge computing, namely **anytime depth estimation** with monocular camera and 2D LiDAR on small robots and mobile devices consisting of resource-limited embedded devices. Consequently, in addition to the depth prediction accuracy, we need to optimize for performance under a **highly dynamic environment**, **limited sensing and computational capabilities**. In fact, all these constraints are very relevant in many common robotic systems, *e.g.*, the Pioneer and K5 Security Robot; they operate in dynamic environments, yet come only equipped with 2D LiDAR and battery-powered embedded computers [2].

As our main contribution, we propose a new metric called *anytime loss*, and optimize an anytime depth estimation model robust to random halts that requires only an energy-efficient 2D LiDAR and

5th Conference on Robot Learning (CoRL 2021), London, UK.

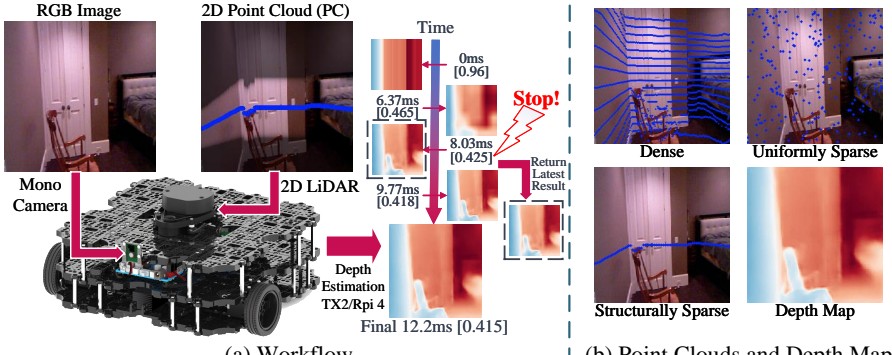

Figure 1: **(a)** Example workflow and timeline of our depth estimation model. As computation progresses, increasingly accurate depth maps are generated (numbers in brackets denote RMSE). Inference can be halted at any time (*e.g.*, when a deadline is reached or a more urgent task needs to start) and the best estimation is returned. **(b)** Point clouds (PC) with different sparsity structures and the output depth map.

a monocular camera. We design the network architecture as a two-stream encoder-decoder with an error correction unit (ECU) placed in middle (Figure 2). We optimize the encoder by utilizing the sparsity in 2D PC shown in Figure 1 yielding a light-weight encoder with marginal accuracy loss. We optimize the decoder by introducing early-exit paths. Both methods are proved to be effective theoretically and practically in improving model's dynamic performance under random halts.

Exhaustive experiments on NYU Depth v2 and KITTI, under both supervised and self-supervised training, show that our method outperforms most state-of-the-art baselines by taking the highly non-uniform depth measurements into consideration with significantly smaller model sizes. Furthermore, without resorting to any quantization and pruning, we show that our model can perform real-time depth estimation on an NVIDIA Jetson TX2 system operating with a computation budget larger than 6.37ms (157FPS) and an energy figure of only 0.2J.

## 2 Related Work

### 2.1 Depth Estimation with Camera and LiDAR

Eigen *et al*. [17] introduce convolution neural networks (CNNs) to the problem of depth estimation. Sparse convolution was then proposed to address the variant sparsity of point clouds [14]. Jaritz *et al*. [18] design an encoder-decoder network to accomplish both depth estimation and semantic segmentation with only the last layer changed. However, these methods simply rely on network architectures to address various structures of point clouds. The performance of a straightforward network drops significantly when the sparsity of point clouds increases [1, 18]. Methods tailored for different sparsity of point clouds are thus proposed. Based on sparsity level of point clouds, camera-LiDAR methods can be divided into three categories: dense, uniformly sparse, and structurally sparse, as illustrated in Figure 1(b). Our work focuses on the most challenging third setting, *i.e.*, depth measurements are sparse and highly non-uniform.

**Dense Point Clouds.** For the dense point clouds in Figure 1(b), the surface normal can serve as a good intermediate feature. More precisely, the density of point clouds ensures that every point has enough data from its neighbourhood; this leads to an accurate estimation of the surface normal. Zhang *et al*. [15] propose to first estimate the surface normal and then combine it with PC for RGB-D depth completion. Qiu *et al*. [13] extend this idea to the camera-LiDAR sensor setting. After the surface normal is estimated, their method generates two depth estimation results with (`image, PC`) and (`surface normal, PC`) taken as inputs, respectively. The final result is obtained by fusing the two depth maps with an attention mechanism.

**Uniformly Sparse Point Clouds.** The uniformly sparse setting shown in Figure 1(b) refers to the scenario where very few points are uniformly scattered across a projected plane. Jaritz *et al*. [18] claim that naively applying CNNs to sparse data can easily fail since CNNs are sensitive to missing data. To address the problem, they propose to use a NASNet encoder [19] to encode the image and sparse depth plane separately. The encoder for the depth plane acts as a depth completion unit to densify the sparse depth map without referring to information from the image plane. Ma *et al*. [1] propose a single regression network to learn directly from raw RGB-D data, and explore the impact

of the number of depth samples on prediction accuracy. Chen *et al.* [5] propose a convolutional spatial propagation network (CSPN) to learn the affinity matrix for depth prediction.

**Structurally Sparse Point Clouds.** The structurally sparse PC illustrated in Figure 1(b) is the most challenging scenario where a few LiDAR points are structurally scattered across a line or a few fixed locations. While this is commonly produced by depth sensors in low-cost devices, such as 2D LiDAR on floor-sweeping robots, little research targets this setting. For instance, [2] and [11] propose to estimate a dense depth map with a 2D LiDAR by stacking the LiDAR points along the gravity direction and then estimate the depth deviation from the generated surface to the actual object surface. However, they rely on the assumption that the depth of an actual object surface is similar to the generated surface, which does not often hold true in open spaces. Our proposed method provides better ways of utilizing sparse 2D points rather than simply stacking them.

## 2.2 Depth Estimation with a Monocular Camera

In recent years, deep learning methods have been considered for depth estimation using a monocular camera due to its low implementation cost. As the general trend has been to rely on deeper and more complicated networks in order to achieve higher accuracy, the state-of-the-art depth estimation algorithms do so at the cost of increased computational complexity. To achieve real-time inference on an embedded system, Wofk *et al.* [20] demonstrate FastDepth, a low-latency, high-throughput monocular depth estimation method that can run on embedded systems. Nekrasov *et al.* [12] propose a method performing semantic segmentation and depth estimation jointly with a single model.

## 2.3 Self-Supervised Depth Estimation

Recent work on self-supervised depth estimation [21–23] shows great potential to effectively make use of unlabeled data. Indeed, the self-supervised methods can achieve comparable or even better performance than supervised ones. Our work focuses on the model itself and addresses to following fundamental question: *Can ones make the depth estimation model satisfy strict power/performance constraints, yet be practical in real-world applications?* In the experimental section, we show that our model can indeed work very well with a state-of-the-art self-supervised training framework.

# 3 Problem Formulation

## 3.1 Depth Estimation with a Monocular Image and 2D Point Clouds (PC)

With an input RGB image and 2D PC, the depth estimation model needs to predict the depth for every image pixel, while yielding a depth map with the same shape as the input image. Depth prediction must finish *before* the model runs out of the computational resources, *e.g.* time and/or energy. The PC generated by a 2D LiDAR consists of a set of points in a plane. We project the PC points to the image plane, and then sequentially concatenate the depth value of these projected points into an 1D vector $C$. An image $I$ is a 3D array with two spatial dimensions. The output depth map $D$ is a 2D matrix with the same spatial shape of the image; each element in $D$ is the predicted depth for the corresponding pixel in the image, the computation budget $B$ is a scalar indicating *e.g.* how much time or energy is available to estimate the depth. In the end, we formulate our depth estimation model as a function $D = f(I, C, B)$.

## 3.2 Anytime Depth Estimation

In robotic systems, due to the dynamic nature of the real world application, the allowable computational budget (*i.e.*, time and/or energy) $B$ varies with time. We cannot know the computational budget in advance and the model has no control over it; to simplify the problem formulation, we assume that the *available* computational budget $B \in (0, B_{max})$ is a random variable and, before every inference, the exact budget $b$ is sampled from it. Intuitively, the objective of an *anytime algorithm* is to get a satisfactory output no matter how much computational budget is available. With the loss function $l(\cdot)$, the overall objective of anytime algorithm is to minimize the anytime loss, i.e., the expected loss under the budget distribution: $L(f) = E_{I,C,B}[l(f(I, C, B))]$.

To make the anytime loss practical, we define the **discrete budget anytime algorithm**, *i.e.*, if there exists a partition $P$ for $(0, B_{max})$ such that $\forall P_i \in P$ and $\forall B_j, B_k \in P_i, f(I, C, B_j) = f(I, C, B_k)$,

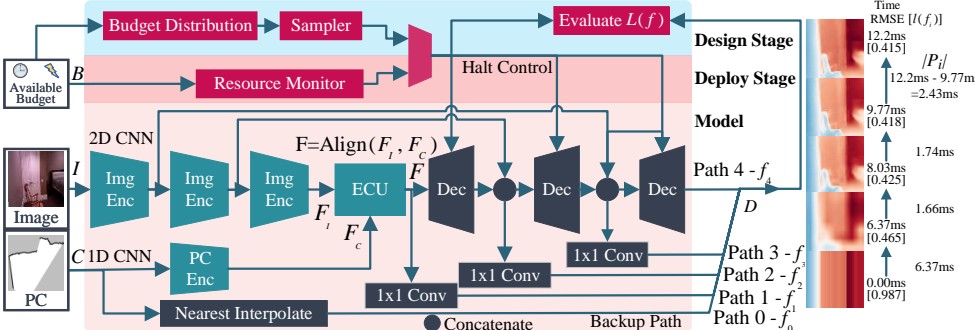

Figure 2: Proposed network architecture. First, two dedicated encoders map the input image $I$ and point clouds $C$ (PC) to hidden features $F_I$ and $F_C$ respectively. Then, one error correction unit (ECU) fuses feature representations from these two modalities. Finally, a shared decoder upsamples the ECU output. At the design stage, we sample from simulated computational budget distributions, *e.g.*, time and energy distributions, halt the inference when the model hits the budget and evaluate the performance with our proposed metric $L(f)$. At the deployment stage, we monitor the system and halt the inference when necessary. Example shown right is annotated with elements for loss $L(f)$ calculation: execution time, accuracy loss RMSE in brackets and length of each interval of each budget partition.

we say the model $f$ is a discrete budget anytime model. Here the $|P_i|$ is the length of the budget interval that corresponds to the partition $i$. Since for any $B \in P_i$ the output of $f(I, C, B)$ is constant, we can define a new function $f_i(I, C) = f(I, C, B)$. The model can be rewritten into: $f(I, C, B) = \sum_{i=1}^{|P|} f_i(I, C) I_{P_i}(B)$, where $I_{P_i}(B)$ is the indicator function, *i.e.*, $I_{P_i}(B) = 1$ iff $B \in P_i$, otherwise $I_{P_i}(B) = 0$. We assume that the computational budget random variable $B$ obeys a uniform distribution $U(0, B_{max})$, then the optimization objective can be rewritten as:

$$
L(f) = E_{I,C,B}[l(f(I, C, B))] = \int_0^{B_{max}} \left\{ \sum_{i=1}^{|P|} [l(f_i(I, C)) I_{P_i}(b)] \text{Prob}(B = b) \right\} \mathrm{d}b
$$

$$
= \frac{1}{B_{max}} \sum_{i=1}^{|P|} l(f_i(I, C)) |P_i|
$$

(1)

where $|P_i|$ is computed for each budget interval across partition $P$. Note that, all these budgets are defined under the assumption that the system configuration is kept the same, *i.e.*, the CPU clock frequency is set to run at the maximum allowable frequency.

## 4 Architecture Design and Optimization

### 4.1 Motivation

The intuition behind our design closely aligns with our problem setting. Consider the discrete budget anytime optimization objective $L(f)$ proposed in the previous section. Here we propose two ways to optimize the objective:

1. For each term $l(f_i(I, C)) |P_i|$, **if** we can minimize the budget required at each stage from $|P_i|$ to $|\hat{P}_i| < |P_i|$ by changing the model from $f_i$ to $\hat{f}_i$, while keeping $l(\hat{f}_i(I, C)) \approx l(f_i(I, C))$, then **it is expected** that $l(\hat{f}_i(I, C))|\hat{P}_i| \leq l(f_i(I, C))|P_i|$ and then $L(f)$ is optimized.

2. **If** we can split each term $l(f_i(I, C))|P_i|$ into sub-terms $l(f_{i_j}(I, C))|P_{i_j}|$ with $\sum_j |P_{i_j}| \approx |P_i|$ but $l(f_{i_j}(I, C)) < l(f_i(I, C))$, then $\sum_j l(f_{i_j}(I, C))|P_{i_j}| \leq l(f_i(I, C))|P_i|$, and $L(f)$ is optimized.

### 4.2 Model Optimization Based on U-Net

We adopt the U-Net [24] style network architecture, shown in Figure 2, where the model $D = f(I, C, B)$ can be partitioned into an encoding part: $F = f_{enc}(I, C, B)$ and a decoding part $D = f_{dec}(F, B)$, in which $F$ is the hidden feature. The encoder and decoder are optimized as discussed above. For the encoder $f_{enc}$, we design a cross-modality encoder with <1.5M parameters by exploiting the low dimensionality in 2D PC and alignment-based data fusion module. For the decoder $f_{dec}$, we generate coarse, but low-latency depth maps on-the-fly by introducing budget-aware light-weight early-exit paths into the model. The details are discussed next.

**Encoder Optimization:** To design a cross-modality encoder, there are two problems to solve: *How to extract relevant features and how to fuse them?* Most previous methods [1, 5, 13, 25, 26] design feature encoders to map the image and PC to a shared feature space with two spatial dimensions; this is because they work with dense or uniformly sparse PC, where both modalities provide information of the global region, as shown in Figure 1(b). In such a case, the feature encoders for both modalities are similar and data fusion is as simple as adding or concatenating features from the two modalities. However, as Figure 1(b) shows, the case of 2D PC is different. Depth measurements in 2D PC only correspond to a limited area of the entire image plane. In problem formulation section, we represent the PC with an 1D vector. So instead of trying to extract a feature with two spatial dimensions based on an 1D vector, we keep the PC feature low-dimensional with only one spatial dimension. Moreover, because the image and PC features have different spatial dimensions now, conventional fusing methodologies are not applicable here; so we also propose an alignment-based feature fusion module to fuse the features from image and PC with different spatial dimensions.

As Figure 2 shows, features $F_I$ and $F_C$ are extracted with two dedicated encoders and fused with an error-correction unit (ECU). Both encoders keep the same spatial dimension before and after encoding, 2D for the image and 1D for the PC. The 1D feature extraction requires much less computation than the 2D feature extraction, so the computation budget is saved and $|P_i|$ is reduced.

ECU fuses $F_I$ and $F_C$ by aligning features at spatially corresponding locations. Points in PC are aligned with the neighboring pixels in the image. We assume this is also true for features extracted from the image and PC. As shown in Figure 3, the ECU extracts the neighboring image features and compares it with the PC features. Based on this comparison, the ECU generates the error compensation value and applies it to the image feature.

**Decoder Optimization:** With a vanilla decoder, before the decoder finishes computation, the best depth map we can get is generated by the PC nearest interpolation. The high loss value together with the large computational budget required by the entire network contributes greatly to the loss $L(f)$.

To reduce the loss when the inference is halted half-way, we export the intermediate features from different stages of the decoder and convert the feature maps to the depth maps with small CNNs. For example, in Figure 2, path 2 extracts the intermediate feature generated by the first-stage decoder, and converts the feature map to a depth map with an $1 \times 1$ convolution layer. Compared with the interpolated depth map, path 2 returns a more accurate depth map; compared with the depth map generated with entire decoder, path 2 requires a smaller computational budget. As the computation progresses, depth maps generated with intermediate features become increasingly more accurate. Here, by using early-exit paths, we introduce a new axis in the quality-speed trade-off space. Considering that the feasible positions for inserting an early-exit path is limited, in practice, we optimize the decoder by manually inserting a few early-exit paths and evaluating on various datasets.

### 4.3 Network Architecture

#### 4.3.1 Light-weight Encoder: Backbone

We design two encoders that map input images and PC into their hidden feature spaces respectively, as shown in Figure 2. While targeting low latency, we consider MobileNet v1 [27] as our encoder of choice. To further reduce inference time and memory requirements, we remove the last 9 computation-heavy convolution layers of one standard MobileNet while keeping the down-scaling factors unaffected, which reduces the parameter count and computation of our image encoder from 2.94M to 1.08M and from 0.582 GMACs to 0.305 GMACs, respectively. Our PC encoder is a fully convolutional network with 1D convolutions. We evaluate the effectiveness of our design in Section 5. The most obvious benefit of our PC encoder design is that the 1D convolution costs very few parameters and computations, with only 44k parameters and 0.966 GMACs.

#### 4.3.2 Light-weight Encoder: Error Correction Unit (ECU)

We train a CNN to perform the feature alignment mentioned in the encoder optimization section. Feature $F_C$ from PC carries accurate information of a limited region. If we align the ambiguous image feature $F_I$ with $F_C$, then a more accurate global depth feature can be expected. As Figure 3 shows, ECU contains two parts: alignment error extraction and alignment error compensation.

**Alignment Error Extraction:** The mismatch between the features $F_C$ and $F_I$ is equivalent to the mismatch between $F_C$ and the spatially corresponding region in $F_I$. The ECU extracts the

spatially corresponding region by cropping a 3-pixel-wide band around the projected positions and then summarizes these regions into a feature with one spatial dimension via convolution. The summarized image region feature has the same spatial dimension with the PC feature now. We extract the alignment error feature by concatenating and passing them through an 1D point-wise convolution layer.

**Alignment Error Compensation:** Given the alignment error feature, we use an 1D convolution layer to generate the compensation value while has only one spatial dimension. We broadcast the compensation to the image feature by nearest interpolation.

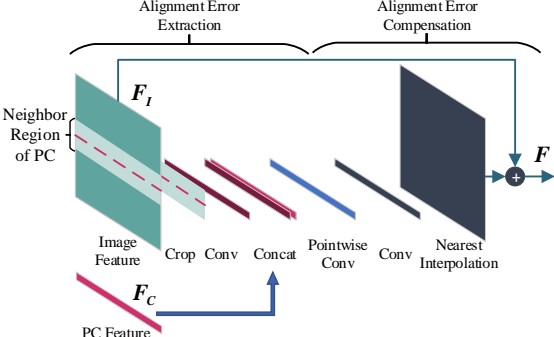

Figure 3: The Error Correction Unit (ECU) design.

#### 4.3.3 Anytime Decoder with Early Exit Paths

We design our anytime decoder based on FastDepth [20]. The decoder is composed of five up-sampling layers, where each up-sampling layer consists of a depth-wise convolution layer and an interpolation layer. After some selected decoder layers, we insert early-exit paths to generate the depth with intermediate decoder features. To make sure that the early-exit paths would not lead to a large computation overhead, the early-exit paths in our decoder contain only one or two point-wise convolution layers. Moreover, compared with FastDepth and considering that the feature from our encoder has a better quality, we reduce the numbers of decoder channels; by doing so, the parameter count of our decoder drops from 0.809M to 0.525M.

## 5 Experimental Results

### 5.1 Experimental Setup

**Datasets.** We set up datasets and accuracy evaluation metrics on NYU Depth v2 [28] and KITTI Odometry [29] datasets following Ma *et al.* [1], except that for the KITTI Odometry dataset, we only use left-view images. To simulate the PC generated by a 2D LiDAR, we sample a line of points at the center of the ground truth depth map.

**Implementation.** The network is implemented in PyTorch and trained with 16-bit float point precision. For training, the batch size is set to 96 and the learning rate is initially set to 0.05 then it decreases to 0.02 with a polynomial learning rate decay scheduler. The optimizer is SGD with an $1 \times 10^{-4}$ weight decay. Finally, the encoder MobileNet v1 is pretrained on ImageNet [30].

**Deployment.** We deploy our network on both Raspberry Pi 4 and NVIDIA Jetson TX2 with TVM [31] and evaluate the speed and energy consumption under random halts. The power of Jetson TX2 and Raspberry Pi 4 is measured on the power supplier with INA226 [32] under 50 Hz sample rate. No additional network pruning or quantization is used in the deployment.

**Baselines.** Every open-sourced baseline is trained with its original training method. We only change the input PC from dense or uniformly sparse PC to 2D PC. For methods compatible with TVM, in order to mimic an anytime algorithm, we change the input image resolutions, fine-tune the model for each resolution and run them in parallel. For a fair comparison, the parameter count and flops reported for each resolution only consider the corresponding model, instead of all of these models.

### 5.2 Results on NYU Depth V2 Dataset

Table 1 presents the quantitative evaluation results in terms of accuracy, model size and inference speed. Figure 4 visualizes the depth maps predicted by each path in our model. Results from different paths are generated in only one run with the same model. For a fair comparison in terms of image resolution, we also evaluate our method with $224 \times 320$ images. As shown in Table 1, no significant difference in accuracy is found.

Compared with most methods in Table 1, our method achieves a comparable or even better precision with an order of magnitude smaller model size. For instance, using only 6.4 ms and 0.3 GFlops, path1 returns a depth map with better quality than EncDecNet [8]. As the computation progresses, depth maps with a better accuracy are predicted. The latest depth map generated by path 4 (at

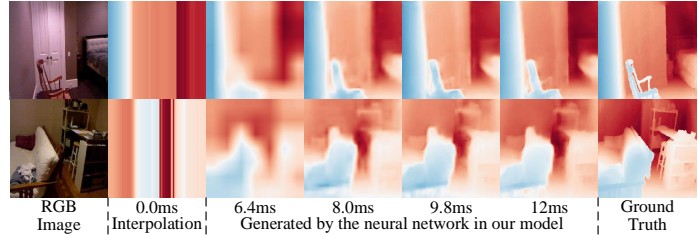

| RGB Image | 0.0ms Interpolation | 6.4ms | 8.0ms | 9.8ms | 12ms | Ground Truth |
| --- | --- | --- | --- | --- | --- | --- |
| | | Generated by the neural network in our model | | | | |

Figure 4: Anytime predictions on NYU Depth v2. The depth maps are generated step-by-step in one run with the same model. The first prediction at 0.0ms is generated with nearest interpolation, if the neural network is halted before making any prediction. Predictions from 6.4ms to 12ms generate more details in the depth map as computation progresses. For more examples, please refer to the Section S.1 of our Supplementary Material.

Table 1: Comparison with prior work in terms of accuracy, latency and network size on the NYU Depth v2 dataset. Latency is measured on the NVIDIA TX2 system. For $\delta_1$, the higher, the better; for other variables, the lower, the better. "Param" is short for parameters count (in millions). "NA" means the method is not open-sourced or not supported by TVM. Our experiment with a different input size is presented in the last row.

| Method | | Input size | Latency↓ | RMSE↓ | Rel↓ | $\delta_1$ ↑ | Param | GFlops |
| --- | --- | --- | --- | --- | --- | --- | --- | --- |
| FastDepth [20] | | 224×224 | 17.1ms | 0.583 | 16.4% | 0.767 | 4.016 | 0.867 |
| Parse [2] | | 256×320 | NA | 0.442 | 10.4% | 0.878 | NA | NA |
| S2D [1] | | 228×304 | 554.7ms | 0.426 | 10.0% | 0.885 | 28.39 | 8.910 |
| EncDecNet [8] | | 228×304 | NA | 0.635 | 15.5% | 0.775 | 0.484 | 1.276 |
| CSPN [5] | | 228×304 | NA | 0.379 | 7.91% | 0.916 | 218.122 | 261.746 |
| **Ours** | Path0 | 224×224 | 0.0ms | 0.9871 | 27.90% | 0.6605 | 0.000 | 0.000 |
| | Path1 | 224×224 | 6.4ms | 0.4651 | 11.52% | 0.8598 | 1.489 | 0.319 |
| | Path2 | 224×224 | 8.0ms | 0.4225 | 10.01% | 0.8871 | 1.983 | 0.423 |
| | Path3 | 224×224 | 9.8ms | 0.4184 | 9.80% | 0.8906 | 2.006 | 0.496 |
| | Path4 | 224×224 | 12.2ms | 0.4156 | 9.80% | 0.8909 | 2.014 | 0.594 |
| **Ours** | Path4 | 224×320 | 16.42ms | 0.4191 | 9.84% | 0.891 | 2.014 | 0.848 |

12.2ms) has a lower RMSE than all baseline method except CSPN. But considering the two orders of magnitude difference in computation counts, our solution is more suitable for edge devices.

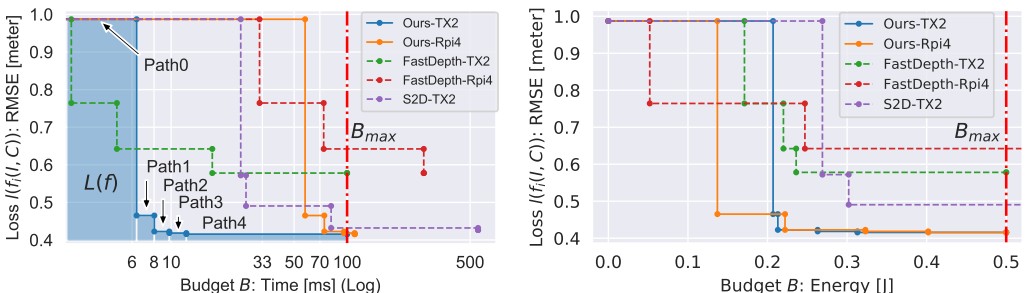

Figure 5: NYU Depth v2 results with different computational budgets. Every drop on the curve corresponds to an optimized prediction made by the model; the area under the curve equals the anytime loss $L(f)$ according to Equation 1. Path 0 generates a backup depth map with the highest RMSE (0.9871) by interpolation, if the neural network is halted before making any prediction, while results on other paths are predicted by neural networks. Predictions requiring more computational budgets than the maximum budget $B_{max}$ are considered to be ineffective predictions. For instance, for S2D with maximum 0.5J energy budget, only paths 0-2 are effective since paths 3-4 require 0.954J and 6.244J respectively (exceeding the maximum energy budget 0.5J). For more detailed experiment results, please refer to the Section S.3 in our Supplementary Materials.

**Anytime Evaluation.** We evaluate the dynamic performance of our model with random computation budgets caused by random interruptions. Evaluations are based on our anytime loss proposed in Equation 1, expected accuracy loss under random available computational budgets. RMSE is used as the accuracy evaluation function $l(\cdot)$ in Equation 1. Two computation budgets, *i.e.*, time and energy, are considered in our experiments. Experimental results are listed in Table 2 and illustrated in Figure 5. Since other baseline methods contain custom operators not compatible with TVM, we only evaluate our method against FastDepth [20] and S2D [1].

Our method outperforms all the baseline methods on NVIDIA Jetson TX2 system, but fails on Raspberry Pi 4 when the maximum budget becomes too small to make a fair comparison. Figure 5 provides the intuition behind these results since every drop on the curve corresponds to an optimized prediction and the area under the curve equals the anytime loss $L(f)$, according to Equation 1.

Table 2: Anytime loss evaluated with time and energy computational budgets. $L_T$@33ms stands for loss $L(f)$ with maximum 33ms as budget. $E$ stands for Energy, "NEE" for No Early-Exit. Bold shows the best results.

| Method | Device | $L_T$@100ms | $L_T$@50ms | $L_T$@33ms | $L_E$@0.5J | $L_E$@0.25J |
|---|---|---|---|---|---|---|
| FastDepth [20] | TX2 | 0.601 | 0.624 | 0.648 | 0.738 | 0.898 |
| S2D [1] | TX2 | 0.604 | 0.740 | 0.869 | 0.763 | 0.987 |
| Ours-NEE | TX2 | 0.484 | 0.554 | 0.626 | 0.773 | 0.891 |
| Ours | TX2 | **0.453** | **0.490** | **0.529** | **0.654** | **0.891** |
| FastDepth [20] | Rpi4 | 0.803 | **0.906** | **0.979** | 0.726 | 0.809 |
| Ours | Rpi4 | **0.755** | 0.987 | 0.987 | **0.582** | **0.746** |

Our first observation is that for most halts, our model returns a depth map with a better quality, *i.e.*, as shown in Figure 5, the curve of our model is always below others; this is achieved by exploiting the sparsity structure in 2D PC and designing low-dimensional encoders. Second, if we remove all the early exit paths in our method, there is a significant increase in the anytime loss, especially when the maximum budget is small ("Ours-NEE" in Table 2). Since for any halt happening before our model makes any prediction, we can only resort to the interpolation based backup path, path 0 which nearly doubles RMSE compared even with the worst result generated by neural networks. Taking experiments on TX2 as an example, by inserting early-exit paths, we reduce the budget required for predicting the first depth map from 12.21ms to 6.37ms, so for nearly half of the halts happening before we finish the entire inference, a better depth map is returned.

## 5.3 Results on KITTI Odometry Dataset

Table 3 compares the performance of our approach with prior work on the KITTI Odometry dataset. We evaluate our model under both supervised and self-supervised training, where self-supervised training is based on MonoDepth2 [21]. Our method achieves a comparable accuracy in terms of relative error and $\delta_1$ score, with orders of magnitude fewer parameters and computations. For more detailed results, see Section S.2 in the Supplementary Materials.

Table 3: Results on KITTI Odometry dataset. First five rows correspond to "Supervised", while last two correspond to "Self-supervised". To save space, we only show the result of Path 4. Bold shows the best results. "Param" is short for parameters count (in millions). "NA" means the method is not supported by TVM.

| Method | Input size | Latency↓ | RMSE↓ | Rel↓ | $\delta_1$ ↑ | Param | GFlops |
|---|---|---|---|---|---|---|---|
| S2D [1] | 228×912 | 374.7ms | 4.272 | 10.0% | 0.902 | 11.49 | 8.593 |
| EncDecNet [8] | 228×912 | NA | 5.462 | 11.6% | 0.839 | **0.484** | 4.000 |
| CSPN [5] | 228×912 | NA | 3.661 | **6.81%** | 0.735 | 218.1 | 261.7 |
| PENet [16] | 352×1216 | NA | **2.694** | 11.4% | **0.942** | 131.9 | 405.3 |
| Ours Path4 | 224×896 | **40.45ms** | 4.183 | 8.06% | 0.891 | 2.014 | **2.375** |
| MonoDepth2 [21] | 192×640 | 77.26ms | **3.259** | 8.16% | 0.926 | 15.24 | 8.051 |
| Ours Path4 | 192×640 | **25.85ms** | 3.608 | **8.13%** | **0.943** | **2.014** | **1.518** |

## 6 Conclusion

In this paper, we have presented a novel anytime depth estimation method on mobile devices, which is robust to random halts and works with power-saving but sensing-capability-limited 2D LiDAR and monocular camera. We formulated our problem as an anytime depth estimation problem and proposed a new metric to comprehensively evaluate its robustness under dynamic interruptions. Based on our proposed metric, we have designed and optimized our model by exploiting the sparsity structure in 2D PC and introducing early-exit paths. Experiments on the NYU Depth v2 and KITTI datasets on edge devices show that, with only 2M parameters, our model has a comparable accuracy and better robustness to random halts comparing with state-of-the-art baseline methods.

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
