# OpenReview forum: "Anytime Depth Estimation with Limited Sensing and Computation Capabilities on Mobile Devices"
_robot-learning.org/CoRL/2021/Conference — CoRL2021 Poster_

### Official Review · Reviewer_Y4hj · 2021-07-18

**Originality:** Very Good
**Technical Quality:** Very Good
**Clarity Of Presentation:** Excellent
**Impact:** 3

**Recommendation:**

Strong Accept: I recommend accepting the paper and will argue for my recommendation even if other reviewers hold a different opinion.

**Summary:**

The paper presents a mono depth completion method based on a single 2D LIDAR scan. The whole model is designed to have a limited computational budget and to exit the inference at different stages depending on the currently available power or computational budget. It shows outstanding results even though the amount of weights and the amount of flops are drastically reduced.

**Issues:**

* Figure 5 should include the 33 and 50 ms mark, to better compare it with Table 2.
* Line 267: should better explain that the Raspberry Pi 4 takes 59 ms to get to the first path, which means that is not fast enough for a fair comparison here

**Reviewer Expertise:**

Good: General knowledge of the area

**Strengths And Weaknesses:**

Strengths:
* Nice and clear presentation of the approach
* Well written paper
* In depth evaluation of the method and comparison on two different relevant dataset
* Introduction on an error-correction unit (ECU), which fuses the 2D features from a mobilenet with the features extracted from a 1D conv-net, by extracting the local neighborhood in the spatial image feature domain and concatenating it with the results from the 1D LIDAR scan features.
* Four way exit approach, which allows to get results after only half the time of execution of the full pipeline
* Impressive results on NYU Depth v2, where after 6.4ms the method performs already 5% better than FastDepth [18], which needs in comparison 17.1ms

Weaknesses:
* Figure 5 should include the 33 and 50 ms mark, to better compare it with Table 2.
* Line 267: should better explain that the Raspberry Pi 4 takes 59 ms to get to the first path, which means that is not fast enough for a fair comparison here


**Summary Of Recommendation:**

The paper was well written and fun to read. I can strongly recommend accepting this paper to CORL. I have a few pointers, which should be improved in a final version, but nothing which should prevent this paper from getting accepted.

---

> ### Author Response · Authors · 2021-08-30
> **Response to Reviewer Y4hj**
>
> - **Comment #1**: Figure 5 should include the 33 and 50 ms mark, to better compare it with Table 2.
>
>   **Revision #1**: Thank you for your advice. We added these two marks in the revised Figure 5.
>
> - **Comment #2**: Line 267: should better explain that the Raspberry Pi 4 takes 59 ms to get to the first path, which means that is not fast enough for a fair comparison here
>
>   **Revision #2**: Yes. Raspberry Pi 4 is not fast enough to make a fair comparison between FastDepth and other methods. We added the explanation in "Anytime Evaluation" subsection of Section 5.2.

---

> > ### Comment · Reviewer_Y4hj · 2021-08-31
> > **Response to response**
> >
> > Thanks for your response. I am still happy with the paper and I am still convinced that this paper should be accepted, in strong contrast to rseL I do not agree that the paper is ill written or hard to understand.

---

### Official Review · Reviewer_HGbf · 2021-07-23

**Originality:** Good
**Technical Quality:** Good
**Clarity Of Presentation:** Very Good
**Impact:** 2

**Recommendation:**

Weak Reject: I recommend rejecting the paper, but will not argue for my recommendation if the majority of other reviewers have a different opinion.

**Summary:**

This paper presents  RGB-D based, computationally efficient, encoder-decoder style depth estimation network with a focus on low power, frequently halting applications.   Unlike other RGB-D fusion architectures, to mimic the limitations existed on small devices, authors use RGB and 2D point cloud as model's input instead of 3D point clouds, and propose an Error Correction Unit, ECU, to algin 2D point cloud and RGB image features. ECU achieves that by extracting point cloud's neighbors in image features and downsizing it to match point cloud feature dimension, and the combined feature is upscaled back to image feature dimension.  Additionally, they incorporate five depth estimation outputs along the computation pipeline in order to achieve anytime depth estimation, where predictions with different levels of detail are available regardless of the halting point of the network.  Authors propose a new metric, anytime loss, which is a weighted sum of loss for each depth output based on everyone's computational time.

**Issues:**

1. Compare the proposed network with and without ECU to demonstrate its effectiveness.
2. add FusionMapping as a baseline
3. try have halt in encoder step and provide the anytime loss for that.
4. clarify the architecture and if not doing downsampling and upsampling, justify the design choice.

**Reviewer Expertise:**

Fair: Some knowledge of the area

**Strengths And Weaknesses:**

Strengths:
1. The idea of anytime depth estimator is novel and useful practically for time-sensitive and low power edge devices.
2. Error correction unit seems like an interesting approach to fuse image and 2D LiDAR features with different dimensions, existing methods commonly use interpolations to expand LiDAR features to image dimension.
3. Comparing results of common models and authors' model on small devices like Jetson and Raspberry Pi with limited time and energy budget is useful for future applications on such edge devices. As many publications focus on SOTA performance on powerful devices, obtaining results on weak devices can guide the community to pay attention to efficiency and ease of deployment of their networks.

Weaknesses:
1. Since all but the first nearest interpolation predictions require the encoder section of the network to be completed, and the first prediction is significantly worse than others, this paper's claim about anytime estimations still requires at least half of the total computation time to yield useful predictions. (In Table 1 path1 takes 6.4ms whereas path4 takes 12.2ms). Would be nice to alter the network architecture to allow it to bypass some layers of encoders to achieve a lower minimum time requirement.
2. Another question I have about architecture is that although authors claim that the network is based on U-net and uses first three conv layers of MobileNetV1 as encoder. Yet, MobileNectV1's first three layers only reduce image dimension by half, and as 1x1 conv layer is used to generate full-size depth estimation. I am not sure if the network downsamples and upsamples features similar to U-net.
3. I think this paper doesn't demonstrate the effectiveness of ECU module, as from the results, Parse, which uses 2D LiDAR by interpolation, performs similarly compared to the proposed network. Would be nice to have a comparison between proposed network with and without ECU module.
4. I think "FusionMapping: Learning Depth Prediction with Monocular Images and 2D Laser Scans" should be included as baseline as it shares a similar architecture with the proposed network and uses 2D LiDAR as input as well.
5. Finally, I have a different opinion about the proposed anytime estimator approach to handle limited time and energy budget for edge devices. The paper considers the halting caused by limited budgets to be a uniformly distributed random variable from 0 to B_mas, and designs the model to give prediction whenever halting happens. However, I would argue that the capturing and processing time for RGBD image can be considered as a constant. Thus, instead of designing a model to perform well under different budgets, constructing a model to fit the given budget seems to be a better option.



**Summary Of Recommendation:**

Overall, this paper shows a novel approach for low power edge device machine learning by handling limited time budgets by anytime depth estimation. The proposed architecture is similar to other RGBD depth estimators with encoder-decoder backbone and point cloud feature fusion, yet the authors introduce multiple branches of prediction between decoder layers to obtain depth estimation with different levels of qualities. Also, a novel anytime loss is introduced to weigh each output's loss by its computational time. Another novel part is the error correction module for RGB and point cloud feature fusion. The paper demonstrates the network's low latency and good performance comparable to existing methods on low power Jetson and Raspberry Pi. However, the newly introduced ECU layer lacks experimental results to demonstrate its effectiveness. And of the claim of anytime prediction, the model design cannot give outputs if halt happens during encoding. Additionally, random halting doesn't match with the situation on low power devices, instead, a fixed time budget seems to be common, which wouldn't have random halting problem and allows a purposely designed model to fully utilize the time budget.

---

> ### Author Response · Authors · 2021-08-30
> **Response to Reviewer HGbf - Part 2**
>
> - **Comment #5**:
>
>   Finally, I have a different opinion about the proposed anytime estimator approach to handle limited time and energy budget for edge devices. The paper considers the halting caused by limited budgets to be a uniformly distributed random variable from 0 to B_mas, and designs the model to give prediction whenever halting happens. However, I would argue that the capturing and processing time for RGBD image can be considered as a constant. Thus, instead of designing a model to perform well under different budgets, constructing a model to fit the given budget seems to be a better option.
>
>   **Response #5**:
>
>   First, let’s assume that all pre-processing procedures would cost a constant budget of say 50ms. Even so, the available inference time can still vary over time. Because the budget the reviewer mentions actually refers to the “wall clock time” instead of the real “run time”, one may never know for how long the OS pauses the model inference to execute other (possibly higher priority) tasks, or for how long the memory accesses stall, or how much time the inter-core communication may take. As the system gets busy, a large portion of time is thus spent on other tasks instead of running the inference; this is especially true for embedded systems with limited computation and memory capabilities.
>
>   Then, as for your assumption, if one uses a USB camera, for instance, one may find that the CPU gets really busy when reading from it. So if the system gets busy, the sensor reading and preprocessing steps may also take a much longer time. We believe that our assumption is therefore well justified, particularly in the context of resource-constrained embedded devices.
>
> - **Comment #6**:
>
>   Overall, this paper shows a novel approach for low power edge device machine learning by handling limited time budgets by anytime depth estimation. The proposed architecture is similar to other RGBD depth estimators with encoder-decoder backbone and point cloud feature fusion, yet the authors introduce multiple branches of prediction between decoder layers to obtain depth estimation with different levels of qualities. Also, a novel anytime loss is introduced to weigh each output's loss by its computational time. Another novel part is the error correction module for RGB and point cloud feature fusion. The paper demonstrates the network's low latency and good performance comparable to existing methods on low power Jetson and Raspberry Pi. However, the newly introduced ECU layer lacks experimental results to demonstrate its effectiveness. And of the claim of anytime prediction, the model design cannot give outputs if halt happens during encoding. Additionally, random halting doesn't match with the situation on low power devices, instead, a fixed time budget seems to be common, which wouldn't have random halting problem and allows a purposely designed model to fully utilize the time budget.
>
>   **Response #6**:
>
>   This is a complex comment so we break it down as follows, for clarity purposes.
>
>   1. **Experiment about ECU layer**: We conducted the new experiment following your suggestion in the third point of the weakness section. The results are reported in Response #3.
>
>   2. **Fixed time budget vs Random halting**: The “fixed time budget” mentioned actually refers to the “fixed wall clock time” instead of the real “fixed run time”; one of our main motivations for introducing the random halting is precisely to simulate the gap between the wall clock time and the real run time. The difference between wall clock time and actual run time can be huge. For example, in the video experiment we provide in the supplementary material, when the system gets busy, the actual run time allocated for the vanilla model is less than 12ms out of 40ms wall clock time. 12ms is the minimum time required for inference, so in video the vanilla model fails to make any prediction.
>
>      In general, one may never know for how much time the OS pauses the model inference to execute other tasks, or for how long the memory accesses stall, or how long the inter-core communication may take. As the system gets busy, a large portion of the time is spent on other things instead of running the inference; this is especially true for embedded systems that have very limited computation capabilities.
>
>   3. Finally, to address the problem regarding the motivation behind random halting, the clarification above (bullet (2) of Response #6) is also made in the Section S.7 of the Supplemental Material.

---

> > ### Comment · Reviewer_HGbf · 2021-08-31
> > **Still not agree with the random time budget arguement**
> >
> > Authors respond my questions about time budget by distinguishing the difference between wall time and cpu run time. And argues that due to halting and multithreading, the actual wall time budget is unpredictable. Which I am not 100% convinced, existing operating systems already provide solutions for applications with a tight time constraint, namely Linux RT kernel. Using that as an example, RT kernel guarantee a maximum latency for interrupt and completely predictable behavior of thread scheduling, so the claim of
> >
> >
> > >one may never know for how long the OS pauses the model inference to execute other (possibly higher priority) tasks, or for how long the memory accesses stall, or how much time the inter-core communication may take. As the system gets busy, a large portion of time is thus spent on other tasks instead of running the inference
> >
> >  is not true. Since this question is closely related to the paper motivation and suitable application, I will not change my recommendation.

---

> > > ### Author Response · Authors · 2021-08-31
> > > **Updated Response to Reviewer HGbf**
> > >
> > > Thank you for your reply. While we understand your comment, we respectfully disagree your view point. In short, RT kernel guarantees the scheduling timing, but cannot guarantee any type of other variations related to: **memory access latency (particularly due to memory hierarchical structure), I/O latency, cross-processor communication latency and last but not least power/thermal consumption variations**. In fact, data reported on the real hardware supports precisely our argument: See page 16 of:
> > >
> > > [Altera 2015] https://elinux.org/images/d/de/Real_Time_Linux_Scheduling_Performance_Comparison.pdf
> > >
> > > as the variation as ~40ms (5th Percentile to 95th Percentile).

---

> ### Author Response · Authors · 2021-08-30
> **Response to Reviewer HGbf - Part 1**
>
> - **Comment #1**:
>
>   Since all but the first nearest interpolation predictions require the encoder section of the network to be completed, and the first prediction is significantly worse than others, this paper's claim about anytime estimations still requires at least half of the total computation time to yield useful predictions. (In Table 1 path1 takes 6.4ms whereas path4 takes 12.2ms). Would be nice to alter the network architecture to allow it to bypass some layers of encoders to achieve a lower minimum time requirement.
>
>   **Response #1**:
>
>   In the present paper, since we use two dedicated encoders for image and PC, respectively, if we add such early-exit paths in the encoder directly, the ECU will need to be invoked for every exit path to fuse them. Even if the ECU has only 370k parameters, it still represents 18% of the parameters of the entire model. Thus the cost invoking ECU multiple times is not negligible so a more sophisticated approach would need to be considered. We will consider finding a systematic approach to insert early-exit paths in the encoder in our future work.
>
>   This is a great suggestion; we really appreciate it.
>
> - **Comment #2**:
>
>   Another question I have about architecture is that although authors claim that the network is based on U-net and uses first three conv layers of MobileNetV1 as encoder. Yet, MobileNectV1's first three layers only reduce image dimension by half, and as 1x1 conv layer is used to generate full-size depth estimation. I am not sure if the network downsamples and upsamples features similar to U-net.
>
>   **Response #2**:
>
>   To clarify, in Figure 2, each image encoder block consists of multiple convolution layers. The reason why we use three blocks to represent the encoder is that two short-cut paths are introduced from the encoder to the decoder. As we describe in Section 4.3.1, we remove the last 9 convolution layers from the MobileNet-V1. To keep the image scale unaffected, we make the down-scaling in the last few layers happen earlier. The spatial dimension of the image encoder output is 7x7.
>
>   **Revision #2**:
>
>   To address this comment, we explicitly point out that we keep the down-scaling factors unaffected in the following sentence in the Section 4.3.1:
>
>   *“To further reduce inference time and memory requirements, we remove the last 9 computation-heavy convolution layers of one standard MobileNet while keeping the down-scaling factors unaffected.”*
>
> - **Comment #3**:
>
>   I think this paper doesn't demonstrate the effectiveness of ECU module, as from the results, Parse, which uses 2D LiDAR by interpolation, performs similarly compared to the proposed network. Would be nice to have a comparison between proposed network with and without ECU module.
>
>   **Response #3**:
>
>   Thank you for your suggestion; to clarify this issue, we conduct a new experiment as follows. We remove ECU and implement our network following the Parse architecture. To compensate for the parameter loss due to the removal of ECU, three MobileNet-V1 convolution blocks (one depthwise convolution layer and one pointwise convolution layer) need to be appended to the encoder. Without the ECU, the model achieves 0.4291 RMSE with 1.942M parameters, while the original model achieves 0.4156 RMSE with 2.014 parameters. Considering that ECU costs only 370k parameters, this shows that the design we propose is effective.
>
> - **Comment #4**:
>
>   I think "FusionMapping: Learning Depth Prediction with Monocular Images and 2D Laser Scans" should be included as baseline as it shares a similar architecture with the proposed network and uses 2D LiDAR as input as well.
>
>   **Response #4**:
>
>   Yes, this paper shares a similar sensor setting as ours, namely monocular camera and 2D LiDAR. However, we find it difficult to use it as a baseline for comparison for two reasons. First, the authors did not release their source code and did not provide any details about the network they use as the backbone (e.g., what exactly is type of ResNet), the parameters used for KNN search, so on, so forth.
>
>   Second, if one looks carefully to the left of Figure 3 in their FusionMapping paper, one can see that the authors are actually using a “band” of points instead of points from a single laser lane. We are not sure if this is for providing a better-looking figure or some other reason.
>
>   To sum up, these two reasons prevented us from considering this prior work as a baseline for our approach.
>
>   **Revision  #4**:
>
>   To address this comment, we cited the FusionMapping paper in “Structurally Sparse Point Clouds” sub-section of Section 2.1.

---

### Official Review · Reviewer_rseL · 2021-07-23

**Originality:** Excellent
**Technical Quality:** Good
**Clarity Of Presentation:** Good
**Impact:** 2

**Recommendation:**

Weak Accept: I recommend accepting the paper, but will not argue for my recommendation if the majority of other reviewers have a different opinion.

**Summary:**

This work proposes an approach for anytime monocular depth estimation on mobile devices when a structured sparse set of measurements in the form of a 2D lidar is available. The authors propose to encode LiDAR scanline measurements using a neural network and splat them to image features that are encoded using a light-weight image network.

**Issues:**

Minor issues

- l. 121 abbreviation "r.v." is never introduced.

- The abbreviation PC is used in almost every sentence in some paragraphs which make these parts dull to read.

Missing citation:

[A] also proposes an efficient depth estimation network that outperforms [18].

[A] Nekrasov et al. Real-time joint semantic segmentation and depth estimation using asymmetric annotation. ICRA 2019

**Reviewer Expertise:**

Good: General knowledge of the area

**Strengths And Weaknesses:**

Strengths:

- Anytime estimation in this setting (e.g. with additional external measurements from LiDAR) is to the best of my knowledge novel.

- Important and undeserved topic. There are only few works in this general direction despite its obvious usefulness for robotics applications.

- Good results in comparison to other real-time approaches.

Weaknesses:

Clarity is an issue throughout the manuscript:

- Section 3.2 states a relatively simple concepts in a bit heavy-handed terms because of the extensive use of mathematical notation. I'd recommend to make this a bit more accessible by removing jargon and explaining with words.

- Section 4.1: similar consideration hold about overly complicated notation, but here I'm not sure if I completely understand the motivation here. For point one: how can you guarantee even in expectation the inequality if the losses are only required to be equal? What happens if the loss for \hat f_i is actully slightly larger on |\hat P_i} is only slightly smaller that |P_i|?

- Paper organization: Section 4.2 talks about network architecture followed by an own section 4.3 that is actually titled "network architecture" but then actually goes on to talk about computational properties. This is confusing. It is not clear what "encoder optimization" means. What is optimization referring to here? The ECU is used as a component and how it qualitatively functions is mentioned already in Section 4.2, but it only introduced in 4.3.2.

- Section 4.3.2 is very hard to understand. What alignment is necessary here? Why is this necessary? How big are the regions? What is the compensation value? It seems that this is just about upsampling the scan line to the full image by extrapolating the scan line in vertical dimension. Why is this so complicated?

- Is a uniform distribution between 0 and B_max well justified? In practice it seems that this would more like by an interval [B_min, B_max] or even a normal distribution.

**Summary Of Recommendation:**

I really like the direction that this paper is taking. Unfortunately, the paper in its current form is extremely hard to read and in its current form not fit for publication. Technical aspects are consequently hard to evaluate in detail, but I believe that the paper in principle is on a good direction. I'm willing to significantly improve my score if the revision appropriately improves clarity.

-- Post rebuttal --

The authors answered my questions and revised the manuscript. I'm consequently raising my score.

---

> ### Author Response · Authors · 2021-08-30
> **Response to Reviewer rseL - Part 2**
>
> - **Comment #4**:
>
>   Section 4.3.2 is very hard to understand. What alignment is necessary here? Why is this necessary? How big are the regions? What is the compensation value? It seems that this is just about upsampling the scan line to the full image by extrapolating the scan line in vertical dimension. Why is this so complicated?
>
>   **Response #4**:
>
>   To address the problem regarding the ECU, we revised Section 4.3.2 to emphasize the motivation behind ECU. Technical details about the size of the neighboring region are also appended to the Section 4.3.2.
>
>   Getting into details, from the perspective of the scan line, we are extrapolating, while from the perspective of the image, we are aligning. The thing is that the monocular image produces an ambiguous depth map, while the scan line provides the ground-truth depth for a limited region. The monocular image ambiguity leads to the misalignment in the shared region, so if we can detect the misalignment in the shared region and compensate for it, we can expect to have better results.
>
>   Following this idea, the first part of ECU extracts the image feature in the shared region (crop), then makes it comparable with the PC feature (conv), and finally compares it with the PC feature (pointwise conv, mimicking pointwise subtraction). Then the second part of ECU corrects the image feature.
>
>   So in general, because the image and PC features are quite different in dimensions, carried information, etc. it takes more effort to fuse them.
>
>   **Revision #4**: To address this comment, we took the following actions:
>
>   - The following sentences are inserted in Section 4.3.2:
>
>     *“Feature $F_C$ from PC carries accurate information of a limited region. If we align the ambiguous image feature $F_I$ with $F_C$, then a more accurate global depth feature can be expected.”*
>
>   - Technical details of the size of neighbor region are added:
>
>     *“The ECU extracts the spatially corresponding region by cropping a 3-pixel-wide band around the projected positions and then summarizes these regions into a feature with one spatial dimension via convolution.”*
>
> - **Comment #5**: Is a uniform distribution between 0 and $B_{max}$ well justified? In practice it seems that this would more like by an interval $[B_{min}, B_{max}]$ or even a normal distribution.
>
>   **Response #5**:
>
>   Zero available computation budget is possible because the preprocessing steps (like image resizing, point cloud transforming) are *not* included in the model; these preprocessing steps may use up all the computation budget when the system gets busy. This is why we need to consider the $[0, B_{max}]$ interval.
>
>   **Revision #5**: To address this problem, the statements above are also included in the Section S.7 of the Supplementary Material.
>
> - **Comment #6**: l. 121 abbreviation "r.v." is never introduced.
>
>   **Revision #6**: Thank you for pointing this issue out. In the revised version, we replace the abbreviation “r.v.” with “random variable”.
>
> - **Comment #7**: The abbreviation PC is used in almost every sentence in some paragraphs which make these parts dull to read.
>
>   **Revision #7**: To address this problem, we reduced the usage of the abbreviation “PC” in Section 2.
>
> - **Comment #8**: Missing citation:
>
>   - [A] also proposes an efficient depth estimation network that outperforms [18].
>   - [A] Nekrasov et al. Real-time joint semantic segmentation and depth estimation using asymmetric annotation. ICRA 2019
>
>   **Revision #8**: In the revised paper, we did include this new reference. Thanks for pointing this out.

---

> ### Author Response · Authors · 2021-08-30
> **Response to Reviewer rseL - Part 1**
>
> - **Comment #1**:
>
>   Section 3.2 states a relatively simple concepts in a bit heavy-handed terms because of the extensive use of mathematical notation. I'd recommend to make this a bit more accessible by removing jargon and explaining with words.
>
>   **Response #1**:
>
>   Thank you for your suggestion. In the revised manuscript, we emphasize the intuition behind the anytime loss to make it easier to understand. In short, the concept of the anytime loss is as simple as the expected loss under a random budget. However, to derive the practical formula (equation (1)) for Neural Networks (NN), some definitions are necessary, e.g., the definition of discrete budget anytime algorithm, which makes the section 3.2 look heavy-handed. To the best of our knowledge this is the first attempt to introduce a quantitative evaluation metric for anytime NN, hence these mathematical derivations are needed to support our narrative.
>
>   **Revision #1**:
>
>   To address this comment, we revised Section 3.2 and the following sentences emphasize the intuition behind the anytime loss:
>
>   - *“Intuitively, the objective of an anytime algorithm is to get a satisfactory output no matter how much computational budget is available.*
>   - *With the loss function $l(\cdot)$, the overall objective of anytime algorithm is to minimize the anytime loss, i.e., the expected loss under the budget distribution: $L(f)=E_{I, C, B}[l(f(I, C, B))]$.”*
>
> - **Comment #2**:
>
>   Section 4.1: similar consideration hold about overly complicated notation, but here I'm not sure if I completely understand the motivation here. For point one: how can you guarantee even in expectation the inequality if the losses are only required to be equal? What happens if the loss for $\hat f_i$ is actully slightly larger on $|\hat P_i|$ is only slightly smaller that $|P_i|$?
>
>   **Response #2**:
>
>   The two bullets in Section 4.1 are not strict derivations; instead, they point out the direction towards an optimized loss. For example, the first bullet says that “IF” we can reduce $|P|$ greatly while keeping the loss close to the original one, then it is expected that the overall loss is reduced.
>
>   Following this direction, we propose a light-weight, yet accurate model. As shown in Figure 5 in the revised paper, by comparing our approach against baseline methods, our model has a much smaller $|P_i|$ and similar loss for $\hat{f}_i$, so consequently, a smaller anytime loss is achieved.
>
>   **Revision #2**: To address this comment, we emphasize the “IF” condition in Section 4.1 explicitly.
>
> - **Comment #3**:
>
>   Paper organization: Section 4.2 talks about network architecture followed by an own section 4.3 that is actually titled "network architecture" but then actually goes on to talk about computational properties. This is confusing. It is not clear what "encoder optimization" means. What is optimization referring to here? The ECU is used as a component and how it qualitatively functions is mentioned already in Section 4.2, but it only introduced in 4.3.2.
>
>   **Response #3**:
>
>   Section 4.2 describes how we optimize the U-Net style architecture, while Section 4.3 describes the model generated by the optimization in Section 4.2.
>
>   ECU is part of the encoder so in 4.2 it’s part of the encoder optimization.
>
>   **Revision #3**:
>
>   To address this comment, we made the following revisions:
>
>   - We changed the title of Section 4.3.1 from *“Light-weight Encoder”* to *“Light-weight Encoder: Backbone”*
>
>   - We changed the title of Section 4.3.2 from *“Error Correction Unit”* to *“Light-weight Encoder: Error Correction Unit”*
>
>   - We changed the title of Section 4.2 from *“Model Optimization”* to *“Model Optimization Based on U-Net”*
>
>   We believe these changes will eliminate any confusion about Sections 4.2 and 4.3 contents.

---

> > ### Comment · Reviewer_rseL · 2021-09-06
> > **Thanks for the response.**
> >
> > Thanks for addressing my questions and comments. Given that clarity improved and that the paper shows an interesting direction, I've increased my score.

---

### Official Review · Reviewer_AwPn · 2021-07-23

**Originality:** Good
**Technical Quality:** Good
**Clarity Of Presentation:** Good
**Impact:** 2

**Recommendation:**

Weak Accept: I recommend accepting the paper, but will not argue for my recommendation if the majority of other reviewers have a different opinion.

**Summary:**

This paper proposes a depth estimation method with 2D lidar measurements and monocular RGB images as inputs, which could perform robustly with sparse and highly non-uniform depth measurements, and give reliable estimations when computation budget is limited.

**Issues:**

1. I have a question about the first way to optimize the objective in section 4.1: As the budget is sampled from the uniform distribution within (0, Bmax), the model should not be able to control it. But it’s said to minimize the budget here, then how does it work? I guess that the value of Bmax gradually decreases during training so that the model is trained to keep a similar performance within lower expected budget. Am I correct?

2. Just a suggestion here: The formula of the loss function used in the optimization objective (equation (1)) is not mentioned. I think it would be helpful to the reader if you could briefly mention about it.


**Reviewer Expertise:**

Fair: Some knowledge of the area

**Strengths And Weaknesses:**

The experimental results on NYU dataset clearly show the advantages of proposed method when working with very sparse depth measurements and low time budget over other state-of-the-art methods,  thus it could establish a new baseline and be a good reference work for relevant research in the future.

This work aims at training a model that provides robust predictions in anytime, hence samples time budget uniformly. I think it would also be interesting to study the actual distribution of interruption time in real applications and applies it to the sampling process, then the model may achieve better expected accuracy.


**Summary Of Recommendation:**

This work targets a practical problem and provides a solution with better performance than other existing methods within the working situation, hence I think it has a high technical value. In addition, the proposed method has a clear and simple framework, thus it’s likely to be embedded to other applications conveniently.

---

> ### Author Response · Authors · 2021-08-30
> **Response to Reviewer AwPn**
>
> - **Comment #1**:
>
>   I have a question about the first way to optimize the objective in section 4.1: As the budget is sampled from the uniform distribution within (0, $B_{max}$), the model should not be able to control it. But it’s said to minimize the budget here, then how does it work? I guess that the value of $B_{max}$ gradually decreases during training so that the model is trained to keep a similar performance within lower expected budget. Am I correct?
>
>   **Response #1**:
>
>   For the first part of your question, yes, the budget refers to the *available budget*, i.e., how much computation can we afford to spend for inference. For the second part, the budget refers to the *required budget*, i.e., how much computation do we actually *need* to finish the inference and get a satisfying result.
>
>   We note that the available budget varies randomly and cannot be controlled by the model. More precisely, the available budget is directly influenced by the environment, while the required budget can be minimized by optimizing the model.
>
>   **Revision #1**:
>
>   To clarify this issue:
>
>   - We included a clarification in Section S.7 of the Supplementary Material.
>
>   - We revised the description of the budget in Section 3.2 as follows:
>
>     *“We cannot know the computational budget in advance and the model has no control over it; to simplify the problem formulation, we assume that the available computational budget $B\in (0, B_{max})$ is a random variable and, before every inference, the exact budget $b$ is sampled from it.”*
>
> - **Comment #2**:
>
>   Just a suggestion here: The formula of the loss function used in the optimization objective (equation (1)) is not mentioned. I think it would be helpful to the reader if you could briefly mention about it.
>
>   **Response #2**:
>
>   We use RMSE as our loss function $l(\cdot)$. Thus $l(\cdot) = ||\hat D - D||^2_2$, where $\hat D$ denotes the ground truth and $||\cdot||_2^2$ denotes 2-norm function. Thus, Equation (1) is extended into:
>
> $$
> \begin{aligned}
> L(f) &= E_{I,C,B}[l(f(I, C, B))] \\\\
> &= \int_{0}^{B_{max}}\{\sum_{i=1}^{|P|}[l(f_i(I, C))I_{P_i}(b)]\text{Prob}(B=b)\}db \\\\
> &= \frac{1}{B_{max}}\sum_{i=1}^{|P|} l(f_i(I, C)) |P_i| \\\\
> &= \frac{1}{B_{max}}\sum_{i=1}^{|P|} ||f_i(I, C) - \hat D||_2^2 |P_i|
> \end{aligned}
> $$

---

### Official Review · Reviewer_2XnV · 2021-07-24

**Originality:** Good
**Technical Quality:** Very Good
**Clarity Of Presentation:** Very Good
**Impact:** 3

**Recommendation:**

Weak Accept: I recommend accepting the paper, but will not argue for my recommendation if the majority of other reviewers have a different opinion.

**Summary:**

The paper addresses the problem of depth estimation using a monocular camera and a 2D LiDAR sensors. The paper proposes an anytime method based on a network architecture that is composed of two dedicated encoders map to extract features from the input of each sensor, a unit to fuse such features, and a decoder to get the final depth map. The proposed method accounts for time/energy budget available to potentially stop the inference and get the corresponding result. Experimental results obtained on public datasets are shown to validate the proposed method.

**Issues:**

The issues that could be addressed to revise the paper are reported in the "Strengths And Weaknesses" textbox.

Other specific comments:
- in practice, it might be difficult to get the line of points at the center of the ground truth depth map, unless the camera and LiDAR are colocated. Thus, taking the line above or below the center would be more realistic.
- Besides just reporting the data, it will be useful to comment on when other methods perform better than the proposed one, e.g., in Table 3.

Minor typos:
- "equiped" -> "equipped"
- "an uniform" -> "a uniform" (because of the consonantal sound)

**Reviewer Expertise:**

Good: General knowledge of the area

**Strengths And Weaknesses:**

Based on existing networks, the paper presents an interesting idea about stopping the inference and explains the motivation behind the choices that resulted in a much smaller network compared to other methods, displaying comparable or better performance in some cases. The paper is also generally well written.

There are two main comments:
- the paper could provide better examples where this stopping is useful for energy purpose: a self-driving car or any moving robot in general typically consumes much more energy from the actuators rather than from the computer. It would be useful to quantify the energy saving.
- the budget model (constant) is quite simplistic and might result in prematurely stopping the inference, as the CPU load might be different at different times. A discussion on a more realistic model would be useful.

**Summary Of Recommendation:**

While the paper could provide a better motivation and discuss a more realistic budget model, the paper presents an interesting anytime method for depth estimation targeted for low-powered systems.

---

> ### Author Response · Authors · 2021-08-30
> **Response to Reviewer 2XnV - Part 2**
>
> - **Comment #3**:
>
>   In practice, it might be difficult to get the line of points at the center of the ground truth depth map, unless the camera and LiDAR are colocated. Thus, taking the line above or below the center would be more realistic.
>
>   **Response #3**:
>
>   True. We set up our experiments based on the existing work (e.g.,[R6]) in order to make comparisons with prior work possible. As such, our experiments show that changing the relative position between the LiDAR and camera does affect the prediction accuracy, as shown in figure of Section S.6 of the Supplementary Material. More precisely, moving the line of points upwards returns a better prediction accuracy (i.e., 0.3655 vs 0.4156 as shown in the Table in Figure R1 of Section S.6 of the Supplementary Material), while moving it downwards results in a lower precision (i.e., 0.4713 vs 0.4156). This is because the points in the upper regions may provide more information about objects in the scene (e.g. tables, chairs, etc.), while the points in the lower regions typically correspond to the ground, hence have less information. Since different robots have different sensor configurations, using a line of points in the center is a sensible choice.
>
>   **Revision #3**:
>
>   To address Reviewer’s comment, we included this new data and the following paragraph in Section S.6 of the Supplementary Material.
>
>   *“We evaluate our model under different camera-LiDAR positions. Results with 0 offset are reported in paper. As the projected position of PC moves upward, the accuracy improves significantly. This is because the lower region typically corresponds to the ground, while the upper region corresponds to the objects in the scene, e.g., tables, walls, etc. Thus, the PC in the upper region may bring more information about the environment to the model, which translates into a better prediction accuracy.”*
>
>   **Reference**:
>
>   [R6] Liao, Y., Huang, L., Wang, Y., Kodagoda, S., Yu, Y., & Liu, Y. (2017, May). Parse geometry from a line: Monocular depth estimation with partial laser observation. In 2017 IEEE international conference on robotics and automation (ICRA) (pp. 5059-5066). IEEE.
>
> - **Comment #4**:
>
>   Besides just reporting the data, it will be useful to comment on when other methods perform better than the proposed one, e.g., in Table 3.
>
>   **Response #4**:
>
>   To save space, we did use a relatively simple form for Table 3 in the initial submission and so we could not provide too many comments on other methods. For the full version of Table 3, please refer to Section S.2 in the Supplementary Material.
>
>   As we can see in this expanded version of Table 3, both CSPN [5] and PENet [14] models use large backbone and post-processing modules. Although these models produce results with an outstanding precision, they are too big to run on real-time embedded devices.
>
>   For the EncDecNet [8], although the model is smaller compared to our model, the required computation is actually much higher than in our case (i.e., 4.0 GFlops for EncDecNet vs 2.0 GFlops in our case); this much higher computational workload for EncDecNet will directly translate into a much higher power consumption compared to our approach.
>
>   **Revision #4**
>
>   To clarify this issue, we included the following sentence in the Supplementary Material (Section S.2):
>
>   *“Table 3 of Section S.2 in the Supplementary Material, shows that our approach uses a model which is precise, yet small and low-power enough to run on real-time embedded devices.”*
>
>   **References**
>
>   [5] X. Cheng, P. Wang, and R. Yang. Depth estimation via affinity learned with convolutional spatial propagation network. In ECCV, pages 103–119, 2018.
>
>   [8] A. Eldesokey, M. Felsberg, and F. S. Khan. Confidence propagation through cnns for guided sparse depth regression. TPAMI, 42(10):2423–2436, 2019.
>
>   [14] M. Hu, S. Wang, B. Li, S. Ning, L. Fan, and X. Gong. Penet: Towards precise and efficient image guided depth completion. arXiv preprint arXiv:2103.00783, 2021.
>
> - **Comment #5**:
>
>   Minor typos: "equiped" -> "equipped"; "an uniform" -> "a uniform" (because of the consonantal sound)
>
>   **Response #5**:
>
>   Thank you for pointing out these typos. We made all these corrections.

---

> > ### Comment · Reviewer_2XnV · 2021-08-31
> > **Thank you for the response**
> >
> > Thanks for the provided detailed response to this review and for providing additional results. Overall, the revised paper improved.
> >
> > The comment on the budget referred to, as written in the previous version of the paper, the assumption that the system power is constant, meaning that the CPU clock frequency is set to run at the maximum allowable frequency, although in practice there might be fluctuations, thus resulting in a potentially conservative strategy in stopping. It might be worth to discuss how this assumption affects the strategy.
> >
> > Also, regarding commenting the results, e.g., Table 3, it is good to comment on the strengths from the other methods besides providing comment just on the proposed method, so that the evaluation is complete.
> > One additional comment as reported in S.2: Path0 for the proposed method reports a latency, param/M, Flops/G of 0, which might be a typo. Please double check.
> >
> > In summary, this review remains on the positive side.

---

> ### Author Response · Authors · 2021-08-30
> **Response to Reviewer 2XnV - Part 1**
>
> - **Comment #1**:
>
>   The paper could provide better examples where this stopping is useful for energy purpose: a self-driving car or any moving robot in general typically consumes much more energy from the actuators rather than from the computer. It would be useful to quantify the energy saving.
>
>   **Response #1**:
>
>   Generally speaking, our paper targets new models and results suitable for small devices like Jetson and Raspberry Pi (or even smaller devices); such devices have tight time and energy budgets as they target future applications on edge/IoT devices. Since most publications focus on SOTA approaches on powerful devices, we believe that providing results on such resource-constrained devices opens up a new research direction for the community with a focus on efficiency and ease of deployment for IoT type of applications.
>
>   Getting into concrete details, our approach targets small robots like nano-drones [R3], mobile devices like mobile phones or AR glasses [R5]. For these devices, the power consumption of the onboard computer is comparable to the power of actuators, especially when the computer runs energy-hungry deep learning (DL) applications. For example, [R3] discusses the energy consumption of a visual-inertial odometry (VIO) system on nano drones. Paper [R4] discusses the SLAM system for energy-budget limited robots. Take the turtle bot 3 as another example in [R1]. Only when the motor stalls, the actuator XL430-W250 reaches its maximum power consumption of 16.8W [R2]. When the motor operates normally, the power consumption of the actuator is much lower (~7W), while the power consumption of Jetson-TX2 can be as high as 15W; so for this class of mobile robots, it’s clearly important to minimize the power that is consumed during computation.
>
>   From a broader perspective, the power consumption is also directly related to the device thermal control, i.e. if the device consumes too much power over a short time interval, the device may overheat and degrade its performance (or even burn in the extreme). So limiting the power consumption budget has a beneficial impact both on packaging (which is important for mobile robots in order to save space/weight) and long-term reliability since operating under higher temperatures does reduce the reliability of various components over time.
>
>   **Revision #1**:
>
>   To address this problem, we emphasize the target application scenario of our approach in Section 1 of the revised paper. The following sentence is revised:
>
>   *“In this work, we argue for a more realistic problem setting for edge computing, namely anytime depth estimation with monocular camera and 2D LiDAR on small robots and mobile devices consisting of resource-limited embedded devices.”*
>
>   **References**:
>
>   [R1] https://emanual.robotis.com/docs/en/platform/turtlebot3/features/#specifications
>
>   [R2] https://www.robotis.us/dynamixel-xl430-w250-t/
>
>   [R3] Suleiman, Amr, et al. "Navion: a fully integrated energy-efficient visual-inertial odometry accelerator for autonomous navigation of nano drones." 2018 IEEE Symposium on VLSI Circuits. IEEE, 2018.
>
>   [R4] J. Tang, B. Yu, S. Liu, Z. Zhang, W. Fang and Y. Zhang, "π-SoC: Heterogeneous SoC Architecture for Visual Inertial SLAM Applications," 2018 IEEE/RSJ International Conference on Intelligent Robots and Systems (IROS), 2018, pp. 8302-8307, doi: 10.1109/IROS.2018.8594181.
>
>   [R5] Hou, T., Ahmadyan, A., Zhang, L., Wei, J., & Grundmann, M. (2020). Mobilepose: Real-time pose estimation for unseen objects with weak shape supervision. arXiv preprint arXiv:2003.03522.
>
> - **Comment #2**:
>
>   The budget model (constant) is quite simplistic and might result in prematurely stopping the inference, as the CPU load might be different at different times. A discussion on a more realistic model would be useful.
>
>   **Response #2**:
>
>   We think there is some misunderstanding going on here: We do *not* assume a constant budget model. As described in Section 3.2, since we cannot know the available budget before the inference completes, we model the available budget as a random variable (r.v.) ranging from 0 to $B_{max}$. In fact, one can see from the paper's main text in Section 3.2 and equation (1) that the budget is a r.v. (not a constant). Furthermore, to evaluate the performance of our model under such a variable budget, we propose a new metric (see equation 1) which considers the expectation of the model's performance under varying (available) budgets.

---

### Meta-Review · Area_Chair_h139 · 2021-09-06

**Recommendation:** Accept (Poster)
**Confidence:** 4

**Metareview:**

The paper presents a mono depth completion method based on a single 2D LIDAR scan.
The paper proposes an anytime method based on a network architecture that is composed of two dedicated encoders map to extract features from the input of each sensor, a unit to fuse such features, and a decoder to get the final depth map. Overall, this paper shows a novel approach for low power edge device machine learning by handling limited time budgets by anytime depth estimation.

All reviewers but one are positive and find the paper addressing an important question and well written/executed with interesting results.
- Clarity on how the budget is used: constant or adaptive? If latter how, and what are the implications of the assumptions on uniform sampling vs say exponential or gaussian distributions.
- Choice and justification for architecture (as pointed out by Reviewer HGbf)

Overall the rebuttal was very useful in clarifying these questions. The authors should ensure that most points from the rebuttal are included in the main paper and if not then should be added to the appendix as FAQs. This will improve comprehension and avoid similar confusion among the readers.

---

### Decision · Program_Chairs · 2021-09-13

**Decision:**

Accept (Poster)

**Comment:**

The paper presents a mono depth completion method based on a single 2D LIDAR scan.
The paper proposes an anytime method based on a network architecture that is composed of two dedicated encoders map to extract features from the input of each sensor, a unit to fuse such features, and a decoder to get the final depth map. Overall, this paper shows a novel approach for low power edge device machine learning by handling limited time budgets by anytime depth estimation.

All reviewers but one are positive and find the paper addressing an important question and well written/executed with interesting results.
- Clarity on how the budget is used: constant or adaptive? If latter how, and what are the implications of the assumptions on uniform sampling vs say exponential or gaussian distributions.
- Choice and justification for architecture (as pointed out by Reviewer HGbf)

Overall the rebuttal was very useful in clarifying these questions. The authors should ensure that most points from the rebuttal are included in the main paper and if not then should be added to the appendix as FAQs. This will improve comprehension and avoid similar confusion among the readers.